# A Pipeline to Investigate Fungal–Fungal Interactions: *Trichoderma* Isolates against Plant-Associated Fungi

**DOI:** 10.3390/jof9040461

**Published:** 2023-04-10

**Authors:** Marianna Dourou, Caterina Anna Maria La Porta

**Affiliations:** 1Department of Environmental Science and Policy, University of Milan, Via Celoria 10, 20133 Milan, Italy; 2Center for Complexity and Biosystems, University of Milan, Via Celoria 16, 20133 Milan, Italy; 3Innovation for Well-Being and Environment (CRC-I-WE), University of Milan, 20122 Milan, Italy

**Keywords:** *Trichoderma*, plant-pathogenic fungi, mutually intermingling growth, antagonism, nucleation dynamic assay

## Abstract

**Featured Application:**

**Nucleation dynamic assay is a concrete strategy to identify microbial behaviours—in particular, interactions of *Trichoderma* fungi with other members of soil microflora.**

**Abstract:**

Soil fungi play essential roles in ecosystems, forming complex interaction networks with bacteria, yeasts, other fungi, or plants. In the framework of biocontrol strategies, *Trichoderma*-based fungicides are at the forefront of research as an alternative to synthetic ones. However, the impact of introducing new microbial strain(s) on the soil microbiome of a habitat is not well-explored. Aiming to identify a quantitative method to explore the complex fungal interactions, we isolated twelve fungi from three Italian vineyards and identified three strains of the *Trichoderma* genus in addition to nine more plant-associated fungi of different genera. Investigating in dual nucleation assay fungal–fungal interactions, we recognised two types of interaction: neutral or antagonistic. All three *Trichoderma* strains displayed a slight inhibitory behaviour against themselves. *Trichoderma* strains showed a mutually intermingling growth with *Aspergillus aculeatus* and *Rhizopus arrhizus* but antagonistic behaviour against the plant pathogens *Alternaria* sp., *Fusarium ramigenum*, and *Botrytis caroliniana*. Yet, in some cases, antagonistic behaviour by *Trichoderma* fungi was also observed against plant-promoting fungi (e.g., *Aspergillus piperis* and *Penicillium oxalicum*). Our study highlights the importance of studying the interactions between fungi, aiming to clarify better the impact of fungal-based biological fungicides in the soil communities, and offers a pipeline for further applications.

## 1. Introduction

Soil microbial communities—mainly consisting of bacteria, yeasts, protozoa, archaea, and fungi—are a complex system that plays a crucial role in regulating the soil ecosystem and, consequently, the health of the plants [1,2]. In this complex picture, soil fungi play multiple roles in soil communities, having high plasticity and adaptability in response to adverse or unfavourable conditions and forming interactions with the plants or other members of the soil microflora [3]. Some fungi regulate the balance of carbon and nutrients thanks to the production of a wide variety of extracellular enzymes used to break down all kinds of organic matter (lignin, cellulose, chitin, proteins, waxes, etc.) into simpler compounds, which can be later decomposed further or used by plants and other microorganisms [4]. Moreover, fungi participating in nitrogen fixation and hormone production can play a role as biological controllers against root pathogens and protect against drought [5,6].

Biocontrol, including the application of microbial agents, is gaining much attention globally as an alternative solution to conventional fungicides in agriculture [7,8]. Among the different microbes used in biological fungicides, there is an increasing interest in fungal species belonging, in particular, to the *Trichoderma* genus. *Trichoderma* fungi are dominant in nearly all soils and foliar environments and have been reported as decomposers or saprophytic microorganisms, while some species have been characterised as mycoparasitic and endophytic [9]. They can control and antagonise plant pathogens through various direct or indirect mechanisms. Specifically, direct mechanisms include mycoparasitism, a typical behaviour of various *Trichoderma* strains, where the antagonist secretes hydrolytic enzymes (e.g., chitinases, glycanases, and proteases) into the environment, degrading the pathogen’s cell wall [10]. In addition, some *Trichoderma* species secrete volatile and non-volatile compounds, such as antibiotics and other secondary metabolites (e.g., heptelidic acid, terpenoid/steroid, and more), which inhibit the pathogen´s growth and prevent the latter from colonising the plants. Hydrolytic enzymes and secondary metabolites released into the environment synergistically affect plant-pathogenic fungi [9,10]. In addition to molecule production, *Trichoderma* fungi also compete with other soil fungi for essential nutrients, space, and attachment sites ([11,12]. On the other hand, indirect mechanisms of the mode of action of *Trichoderma* fungi include root colonisation, enhancement of plant and root growth, and effective induction of plant defence against various pathogenic fungi. Notably, *Trichoderma* belong to natural-plant-growth-promoting fungi, since they colonise the roots through penetration, utilise compounds released by the host plant, and promote plant growth and photosynthetic rate. Furthermore, *Trichoderma* release metabolites and trigger the production of antimicrobial compounds by the host plant, inducing resistance against plant pathogens [10]. The interaction between *Trichoderma* fungi and plants is characterised as mutually beneficial [9]. Thanks to these properties, they are often used as biocontrol agents against plant fungal diseases [13]).

However, the introduction of new strains of microorganisms or microbial products into an agricultural area will have an unknown long-term impact on the microbiome and the biodynamics of the treated area. Although the molecular mechanisms of the complex interactions between plant-promoting and plant-pathogenic fungi have been extensively investigated and revealed useful findings (e.g., the discovery of genes encoding cell-wall-degrading enzymes) [14], it is increasingly evident that the understanding of biocontrol in its overall mode of action is incomplete. The interaction of *Trichoderma* fungi with other fungi naturally present in soil communities from the perspective of nucleation dynamics has been poorly studied, and the published investigations focused only on interactions with plant pathogens [15,16,17,18] and not with other plant-associated fungi.

Among the various agricultural sectors, wine production is one of the oldest from a historical point of view and has been an ongoing agricultural practice for centuries with a high financial impact worldwide. The global vineyard surface area was estimated to be 7.3 million hectares (mha) in 2021, representing about 40% of the world’s cultivated area [19], and Europe is the world’s largest wine-producing continent. To maintain stable production yields and protect crops from pathogenic fungi, a significant part of the wine industry still relies on fungicides, while some owners have adopted environmentally friendly agricultural practices. Nevertheless, according to Regulation (EU) 2018/848 of the European Parliament and of the Council, farmers are allowed to use preparations of microbial-origin products or microorganisms to improve the overall condition of the crop [20]. Since vineyards are at the forefront of using organic farming practices, they represent a great ecosystem for identifying new strategies to control the growth of pathogenic fungi, thus helping to limit the use of chemicals.

In the present study, we investigate the interaction of three *Trichoderma* strains with each other and with nine plant-associated fungi isolated from vineyards in northern Italy using a dual nucleation dynamic assay. This study offers a pipeline to investigate the effect of the coexistence of different fungal strains, with vineyards representing only an example of application.

## 2. Materials and Methods

### 2.1. Soils Samples Collection from Vineyards Located in Northern Italy

Three vineyards distributed in the Northern part of Italy were selected for this research. Two of the three are located in the Lombardy region (Vigna Grande is located in Via Giuseppe Mazzini, 20, Botticino BS, Brescia, Italy; Vigne Olcru is located in Via Buca 26/a, Santa Maria della Versa, Pavia, Italy). In those two vineyards, conventional farming practices are used to cultivate the grape plants. The third is located in Piemonte (La Raia, Località Lomellina 26, Gavi, Alessandria, Italy) using organic farming practices for grape plant cultivation. From each vineyard, at least four environmental samples were collected during the winter of 2021, and the spring and summer of 2022. A total of 10 samples were analysed, as reported in Table 1. In addition to soil samples from the rhizosphere, dry leaves were also collected from the ground, as fungi can overwinter in diseased plant debris and persist in the soil or debris for a long time.

### 2.2. Isolation of Plant-Associated Fungi

A part of the samples listed in Table 1 was grounded in a mortar containing 30 mL of sterile deionised water. The homogenate was sequentially passed through a sterile nylon filter with pore sizes of 100, 60, and 30 μm (Nylon Net Filters, Merck Millipore Ltd, Tullagreen, Ireland). The material retained on the 30 μm mesh filter was resuspended in 10 mL of sterile $dH_{2}O$. Finally, the last filtrate and the suspension in $dH_{2}O$, generating two sub-samples per sample (named a and b), were checked under an optical microscope (40X magnification) for the presence of spores. *Vitis vinifera* leaves were selected for the isolation of plant-associated fungi (for details on the plants’ growth conditions, see Section 2.4). The leaf’s surface was sterilised with 70% (*v*/*v*) ethanol (Sigma-Aldrich, Steinheim, Germany) and then quickly rinsed in sterile $dH_{2}O$. Leaves were placed on wet filter paper (by 1 mL of sterile $dH_{2}O$) and on 0.8% sterile agar gel (Sigma-Aldrich) in Petri dishes with their abaxial surface upwards. Of the samples generated, as mentioned above, 200 μλ were used for the inoculation of the leaves. Petri dishes were closed and sealed with parafilm to preserve moisture and then incubated for 5 to 7 days in a growth chamber (T = 18/22 ± 1 °C (night/day), humidity ≥ 80%, intensity = 25 μmol × $m{-}^{2}$ × $s{-}^{1}$, photo-period = 16 h light: 8 h dark).

After the incubation, different fungi able to grow on the leaf’s surface were manually selected and aseptically placed in Petri dishes on Sabouraud 4% dextrose agar medium (SDA, Merck kGaA, Darmstadt, Germany) containing 0.1 mg/mL ampicillin (Sigma-Aldrich) to suppress bacterial growth and incubated at 27 ± 1 °C for 7 days in a static incubator (Memmert GmbH ^+^ Co. KG, Schwabach, Germany). Fungal colonies with morphological differences were regularly sub-cultured on fresh plates of potato dextrose agar (PDA, Merck kGaA) and incubated at 27 ± 1 °C for different time periods depending on the fungal growth rate to ensure the isolation of pure cultures. Finally, pure cultures were obtained and maintained on PDA at 4 °C.

### 2.3. Morphological and Molecular Identification of the Isolated Fungi

The fungal isolates were identified at the genus level using cultural and morphological features (e.g., colony growth pattern, conidial morphology, and pigmentation).

Molecular identification was achieved by phylogenetic analysis of the internal transcribed spacer (ITS rDNA) region, which is hypervariable as there is little selective pressure on it, meaning that ITS sequencing gives an extremely high taxonomic resolution for fungi. Genomic DNA of each strain was extracted using the Fungi/Yeast Genomic DNA Isolation Kit by Norgen Biotek (Cat. 27300, Norgen Biotek Corp, Thorold, ON, Canada) according to the manufacturer’s instruction. Briefly, fungal spores and mycelium were harvested from cultures growing on Petri dishes by adding 5 mL of sterile 0.9% (*w*/*v*) NaCl (Sigma-Aldrich) into the plate and gently scraping the surface of the culture. Then, the lysate of each sample was prepared as follows: 1 mL of the mixture was centrifuged at 14,000× *g* for 1 min (Eppendorf Centrifuge 5804 R, Eppendorf, Hamburg, Germany); after discarding the supernatant, the cell pellet was resuspended on the lysis buffer provided by the manufacturer. The mixture was transferred into a Bead Tube, vortexed for 5 min, incubated at 65 °C for 10 min, and centrifuged at 14,000× *g* for 2 min. The supernatant was carefully transferred to a new Dnase-free tube, and the addition of an equal volume of pure ethanol (Sigma-Aldrich) and 300 μL of solution BX was conducted. Following, nucleic acids were bound to the spin column and washed two times. Firstly, 650 μL of the lysate was applied onto the column and centrifuged at 10,000× *g* for 1 min. Then, the column was washed twice by applying 500 μL of wash solution (containing pure ethanol) and centrifuged at 10,000× *g* for 1 min each time. Finally, the eluate was collected by adding 100 μL of the elution buffer to the column and centrifuged at 10,000× *g* for 2 min. The samples were stored at −80 °C until further analysis.

For molecular identification of the fungus, next-generation sequencing experiments, comprising quality control and primary bioinformatics analysis, were performed by Genomix4life S.R.L. (Baronissi, Salerno, Italy). Briefly, DNA quality control was performed using a NanoDropOne spectrophotometer (Thermo Scientific, Waltham, MA, USA) and Qubit Fluorometer 4.0 (Invitrogen Co., Carlsbad, CA, USA). ITS amplification was performed with the primers ITS3–ITS4 (e.g., ITS3f 5’-GCATCGATGAAGAACGCAGC-3′ and ITS4r 5′-TCCTCC-GCTTATTGATATGC-3′). Each PCR reaction was assembled according to Metagenomic Sequencing Library Preparation (Illumina, San Diego, CA, USA). A negative control is included in the workflow; it consists of all reagents used during sample processing (ITS amplification and library preparation) apart from a sample to ensure any contamination. Libraries were quantified using a Qubit fluorometer (Invitrogen Co.) and pooled to an equimolar amount of each index-tagged sample to a final concentration of 4 nM, including the Phix Control Library. Pooled samples were subject to cluster generation and sequenced on the MiSeq platform (Illumina) in a 2 × 250 paired-end format. Data analysis was performed with the Basespace 16S Metagenomics pipeline (Illumina). The UNITE Fungal ITS Database v7.2 is based on FASTA from UNITE Community (2017): UNITE general FASTA release, version 01.12.2017. UNITE Community [21] includes singletons set as RefS (in dynamic files).

### 2.4. Growth Conditions of *Vitis vinifera* Plants

Cuttings with two buds from plants (cultivar *Pinot Nero*) were provided by the Vigne Olcru vineyard (Santa Maria Della Versa, Pavia, Italy) in December 2021. The cuttings were disinfected with 70% (*v*/*v*) ethanol and left to sprout in the greenhouse under controlled conditions (T = 18/23 ± 1 °C (night/day), humidity ≥80%, photo-period = 16 h light : 8 h dark). When needed, young, healthy leaves were collected from the stem of the grapevine plant.

### 2.5. Nucleation Dynamic Assay

Nucleation dynamic assay (single or dual) was used to study the ability of the *Trichoderma* strains to inhibit the growth of plant-associated and plant-pathogenic fungi in 90 mm × 15 mm Petri dishes (Sigma-Aldrich). PDA plates were inoculated with 7 mm diameter mycelial plugs of the active growth stage *Trichoderma* strains, placed 10 mm from the edge of the plate. Mycelial plugs, 7 mm of each tested fungus, were placed separately, opposite to *Trichoderma* stains, 25 mm away from each of *Trichoderma* discs. Control plates were single-inoculated with the *Trichoderma* strains, while *Trichoderma* strains were also tested against each other in a dual assay. The plates were incubated in a static incubator (Memmert GmbH ^+^ Co. KGat) at 27 ±1 °C for 6 days (i.e., 144 h). All cultures were performed in duplicate.

### 2.6. Macroscopic and Microscopic Observations of the Fungal Isolates

The macroscopic morphology of the individual colonies of fungi growing on PDA Petri dishes was described by eye observations, using standard terminology to define common fungal colony types.

Macroscopic pictures of the fungal cultures and the interface between the fungi were captured using an HDR triple camera with the following specifications: (1) 108 MP, f/1.9-aperture lens, 1/1.52-inch sensor with 0.7 μm pixel size, PDAF; (2) 8 MP, f/3.4, 126 mm, 1.0 μm, 5X magnification, PDAF, OIS; (3) 16 MP, f/2.2, 13 mm, 119°, 1.0 μm. Images were acquired every 24 h over a 144 h period. Time-lapse videos showing fungal growth over time were performed by combining the pictures mentioned above using the Clideo program.

Mycelium and spores morphology were observed microscopically using a Leica microsystem optical microscope (Leica microsystems CMS GmbH, Wetzlar, Germany; 40X magnification). Slides for microscopic examination were prepared using material from the margin of each fungal colony, where spores are actively produced. Images were acquired using the Leica LAS X program (Version 3.7.5.24914).

## 3. Results and Discussion

### 3.1. Identification of the Isolated Fungal Strains from Vineyards in Northern Italy

Soil fungi are among the most abundant and diverse taxonomic groups on Earth, playing essential roles in soil ecosystems and contributing to retaining healthy plants [4,5,6,22,23]. In this research, we identified twelve plant-associated fungal species isolated from soil and dry leaf samples obtained from three vineyards in Northern Italy, as reported in Table 1. Fungal strains were identified using both morphological features and genome sequencing (for more details, see the Materials and Method section). The strains have also been deposed and preserved in the Ex Culture Collection (EXF) of the Department of Biology (Biotechnical Faculty, University of Ljubljana (Ljubljana, Slovenia), Infrastructural Centre Mycosmo, MRIC UL) (Table 2). The colony morphology in a Petri dish and the microscopic appearance of these strains growing on PDA plates are shown in Figure 1.

Three different *Trichoderma* stains have been isolated, two of them resulted as unidentified at the level of species (Nos. 2 and 3 in Table 2). Specifically, the *Trichoderma* sp. EXF-17016 (No. 2) remained unidentified at 95.4%, probably because of the lack of a database for comparison due to the high geographical distribution of the genus. On the contrary, *Trichoderma* sp. EXF-17020 (No. 3) was found to be closely related to *T. yunnanense* by 41.3% and 52.6% unidentified. Most of the *Trichoderma* species are dominant in the mycobiome of various soil ecosystems [5] and can quickly adapt and thrive in various environmental situations [24]. They produce a wide array of enzymes and are resistant to many toxic chemicals, including fungicides (e.g., azoxystrobin; 3,4-dichloroaniline; trifloxystrobin; and trinitrotoluene), herbicides, and other organic pollutants [10,24]. The isolated *Trichoderma* species of this study, when grown on PDA medium, initially formed a white or transparent mycelium; within a week, their mycelium turned to several colours (Figure 1). For all three strains, the colony texture is cottony. Highly branched conidiophores and clusters of divergent, usually asymmetrically bent, enlarged in the middle, or cylindrical phialides characterise their microscopic appearance (Figure 1).

We also identified two fungi belonging to the *Aspergillus* genus, which comprises a diverse group with a global distribution in terrestrial habitats. The isolate *A. aculeatus* (No. 4) is usually isolated from rotting fruits and soil, and it has been reported as a plant pathogen of tomatoes and grapes [25]. In addition, *A. aculeatus* produces several enzymes, including cellulases, hemicellulases, pectinases, and proteases, which are used in the biodegradation processes of lignocellulosic biomass [26,27]. On the contrary, the isolate *A. piperis* (No. 11) has recently been described as a new antagonist to plant-pathogenic fungi thanks to its strong antifungal activity [28,29]. Both *Aspergillus* isolates present black-coloured colonies (Figure 1) with a hairy texture. Their hyphae, under the microscope, appear septate, and the conidiophore is enlarged at the tip, forming a swollen vesicle covered with flask-shaped phialides.

We also identified one fungus as *Alternaria* sp., unclassified at the species level by 98.1%. The genus of *Alternaria* is a large and complex group with a global distribution in soil and air. It consists mainly of saprophytic and plant-pathogenic species, which cause pre- and post-harvest damage to more than 4000 host agricultural products [30]. *Alternaria* species are generally considered necrotrophic pathogens, killing the host tissue using cell-wall-degrading enzymes and secondary metabolites (i.e., toxins) [31]. The colour colony of the isolate *Alternaria* sp. EXF-17017 (No. 6) grown on the PDA medium is brown, with a woolly texture and a thin, white margin (Figure 1). Microscopically, the conidia are dark brown with both longitudinal and transverse septa.

One of the isolated fungi has been identified as *F. ramigenum*. The *Fusarium* genus contains numerous species associated with plants as pathogens, endophytes, or saprophytes, while some may cause opportunistic mycotoxicosis in domestic animals and humans [32]. Growing on PDA medium, *F. ramigenum* EXF-17018 (No. 7) produces pale rose colonies with a cottony texture (Figure 1). Unlike the majority of *Fusarium* species, our strain did not produce any distinguishing odour or secreted pigment on the PDA medium. Microscopically, the hyphae of *Fusarium* species are hyaline and septate. Conidia produced by our strain are elongated and aseptate, and do not create linear chains (Figure 1).

The genus *Rhizopus* is probably the best-known genus in the class of Mucoromycetes fungi (belonging to the newly formalised phylum of Mucoromycota), containing species that occur naturally in soils and live on decaying plant material or as airborne spores. Species of *Rhizopus* genus exhibit a complex metabolism and produce a variety of enzymes that either allow them to utilise a wide range of substrates or are used in industrial applications [33]. Species of the genus *Rhizopus* are very fast-growing, producing white mycelium and black sporangia, visible to the naked eye, and with a cottony texture, such as the *R. arrhizus* EXF-17019 (No. 8) isolate (Figure 1). In microscopic observations, the hyphae of *R. arrhizus* are nonseptate, forming rhizoids at the base of the sporangiophores and columella in the sporangium.

Species of *Botrytis* genus are non-obligate plant pathogens and saprophytes with a necrotrophic lifestyle on more than 200 crop hosts and forest trees worldwide [34]. The isolate *B. caroliniana* has been identified as responsible for causing the grey mould of strawberry, raspberry, and blackberry in South Carolina [35]. To our knowledge, this is the first report of the presence of *B. caroliniana* in Europe. Other species, such as *B. fabiopsis* or the new species *B. euroamericana*, have been observed in Europe; however, information on their geographical presence is scarce [36,37]. *B. caroliniana* EXF-17025 (No. 9) grown on PDA medium has a brownish colour with a woolly texture (Figure 1). Microscopically, the hyphae are septate, and the fungus propagates with round to elliptical conidia.

Finally, we isolated and identified three strains belonging to the *Penicillium* genus: *P. copticola*, *P. oxalicum*, and *Peniccilium* sp. Species of the *Penicillium* genus constitute an important part of the soil mycobiome and are of major importance in food and drug production. Our isolate *P. copticola* (No. 10) has been reported as an endophytic fungus of the medical plant *Cannabis sativa L.* (marijuana) [38], exhibiting antifungal activity against plant-pathogenic fungi. Furthermore, the isolate *P. oxalicum* (No. 12) has recently been used successfully as a plant growth promoter and protector against downy mildew disease [39]. Even though the *Penicillium* genus is well-defined and can be easily recognised, species identification is still challenging; so, the last isolate, *Penicillium* sp. EXF-17021 (No. 5), remained unidentified by 75.5% at the level of species. The growth of the three *Penicillium* strains on PDA medium is rugose, and their colony texture is velvety. The colony colour of *Penicillum* sp. is blue-green, *P. copticola* is light grey, and *P. oxalicum* is grey–green (Figure 1). Contrary to other *Penicillium* strains, we did not observe the presence of soluble pigments or colony reverse colours. Microscopically, their mycelium has highly branched networks of septate hyphae. Conidiophores are at the end of each branch, accompanied by spherical conidia.

To sum up, by examining three vineyards located in different geographical areas in Northern Italy, and using a specific protocol, we isolated twelve plant-associated fungi as members of the soil microbe of the examined agrarian landscapes. Based on their role, we can categorise them into two main groups: (1) antagonistic to plant pathogens or plant-growth-promoting fungi, consisting of *T. simmonsii*, two strains of *Trichoderma* sp., *A. piperis*, *P. copticola*, and *P. oxalicum*; (2) plant pathogens, consisting of *A. aculeatus*, *Alternaria* sp., *F. ramigenum*, and *B. caroliniana*. We also identified an opportunistic harmful fungus to animals, *R. arrhizus*.

### 3.2. Fungal-Fungal Interactions Using Single or Dual Nucleation Dynamic Assay

Soil microorganisms naturally interact, forming different relationships that may be inhibitory, stimulatory, mutualistic, or neutral for each partner participating. In addition, the type of interaction also depends on the species involved, not just on the genus they belong to [40]. However, several factors may disturb the population of a group of microbes; thus, their natural balance may be compromised. Such a factor may be the introduction of new microbial strain(s) into a habitat (i.e., an agricultural area) under the concept of biocontrol strategies as an alternative to synthetic fungicides for plant diseases.

In the last few years, many microbial species have been implemented as biocontrol agents in biological fungicides [41]. Among them, thanks to their antagonistic properties against plant-pathogenic fungi, *Trichoderma* strains have received increasing interest. At least 77 commercial *Trichoderma*-based fungicides are available on the global market for use in greenhouses and on open-field crops, containing at least 36 different strains belonging to 13 species of the *Trichoderma* genus [42]. Regarding fungal infections of grape plants, strains such as *T. asperellum* and *T. gamsii* have been successfully examined as a prevention treatment in the form of a commercial mixture against esca infections in Italian vineyards [43]. Trying to elucidate the action of *Trichoderma* strains with respect to plants, recently, Kamble et al. [44] reported that *T. harzianum* enhanced the defence response of grape plants against downy mildew disease by promoting the accumulation of lignin, callose, and hydrogen peroxide and up-regulating defence enzymes (i.e., phenylalanine ammonia-lyase, peroxidase, and 1,3-glucanase).

However, the long-term impact caused by the use of *Trichoderma*-based fungicides on the soil communities has been poorly studied. Savazzini et al. [45], evaluating the fate of *T. atroviride* over time in Italian vineyards, demonstrated that the fungus spread rapidly up to 4 m from the initial inoculation point, and was still present in the treated areas one year after inoculation. The adaptability and stable persistence shown by *Trichoderma* strains to a range of environmental conditions makes urgent the need to study their interaction with members of the soil microflora. Herein, we investigated the interaction between *Trichoderma* isolates from vineyards in Northern Italy with each other and with other plant-associated fungi in vitro, using a nucleation dynamic assay.

The growth of the three *Trichoderma* strains alone (single nucleation dynamic assay) shows that the growing mycelium occupied all of the available space (Figure 2a, and Appendix A) 96 h post-cultivation and then continued to age up to 144 h.

In contrast, the mycelium of the three *Trichoderma* strains co-cultivated in dual nucleation assay against the same fungal partner (Figure 2b: 1 vs. 1, 2 vs. 2, and 3 vs. 3; Appendix A) or against a different strain of the same genus (Figure 2b: 1 vs. 2, 2 vs. 3, and 3 vs. 2; Appendix A) grows as mutual space occupying half of the Petri dish in the same time period. This behaviour has been described as a mutual slight inhibition by Porter et al. [46]. It seems that fungi have developed mechanisms to accurately recognise themselves and not harm cells belonging to the same species or genus; thus, in our case, they achieved equilibrium in terms of space and nutrients from the first days of their co-culture. This study is important because, as previously mentioned, *Trichoderma* fungi are members of the soil microflora, while the impact of the use of *Trichoderma*-based biological fungicides on endogenous *Trichoderma* populations has not been examined.

We then investigated the behaviour of the three *Trichoderma* strains with respect to the other nine fungal strains isolated from the three vineyards by the aim of the dual nucleation dynamic assay. Appendix A shows all combinations at 72 and 144 h post-cultivation. It is possible to summarise two main types of behaviours: neutrality and antagonism/inhibition. According to Porter et al. [46], inhibitory behaviour of the in vitro growth on a solid medium of one fungus against another can be categorised into (a) mutually slight inhibition, (b) growth around, (c) overgrowing, and (d) inhibition at a distance.

Neutrality or mutually intermingling growth is observed between all *Trichoderma* strains with the plant pathogen *A. aculeatus* and the opportunistic harm to animals with *R. arrhizus*, where the mycelia of the two colonies touch each other at the interface and grow together [46,47]. As an example of this behaviour, in Figure 3 and Appendix A, we report the case of *T. simmonsii* with *R. arrhizus* (i.e., 1 vs. 8). Neither partner is inhibited by the presence of the other. *Trichoderma* strain grew under the fast-growing mycelium of *R. arrhizus*; finally, after 144 h of co-cultivation, the mycelium from both partners are mixed, changing the morphology in terms of colour and texture of the co-culture.

On the other hand, interestingly, *Trichoderma* strains showed antagonistic behaviour not only against themselves (as shown in Figure 2) but also against the majority of plant-pathogenic fungi isolated in this research, namely, *Alternaria* sp., *F. ramigenum*, and *B. caroliniana* (the exception being *A. aculeatus* as previously reported), as well as against other fungi, which are reported to be antagonistic to plant pathogens (e.g., *P. oxalicum*) (Appendix A). Similarly, Hammad et al. [15] reported the inhibitory behaviour of *Trichoderma* isolates against the plant pathogen *B. cinerea* in dual cultures.

In the case of mutual slight inhibition, observed between *Trichoderma* strains, it is possible to observe that the growing mycelium produced by both colonies approach each other until they are almost in contact, leaving a well-marked, narrow, unoccupied space between them (e.g., 2 vs. 3 in Figure 3). It is, however, possible to identify two more sub-types of this antagonistic/inhibitor behaviour according to Porter et al. [46], namely, growth around and growth above.

In the case of growth around (case 2 of antagonistic behaviour in Figure 3a), the growth of the second partner (e.g., *F. ramigenum*) is inhibited at the connection point, and its mycelium is forced to grow towards the side of the Petri dish, away from the *Trichoderma* partner (i.e., 3 vs. 7 in Figure 3a and Appendix A). Furthermore, looking at the interface (Figure 3b), it is visible that the mycelia of the two strains are not in contact but they leave an unoccupied space between them, a behaviour observed even after more than 2 weeks. This sub-type of antagonistic behaviour was also identified during the co-cultivation of other *Trichoderma* strains (i.e., *T. brevicompactum*, *T. hamatum*, *T. harzianum*, and *T. koningiopsis*) against *F. oxysporum*, a pathogen of onion plants [16].

We also observed the sub-type behaviour named growth above ([46]) between *Trichoderma* strains and *Alternaria* sp., *P. copticola*, *Penicillium* sp., and *B. caroliniana* (as an example, we show 2 vs. 9 in Figure 3 and Appendix A), where the new-born mycelium of the *Trichoderma* strains grow filamentary above the *B. caroliniana* colony. In this sub-type of antagonistic behaviour, the growth of the second partner is strongly inhibited.

Moreover, for none of our fungi have we seen an inhibition at a distance behaviour, as reported by Porter et al. [46].

Finally, we want to emphasise that despite belonging to the same genus, we have seen differences in the behaviours of the three *Trichoderma* strains. For instance, although *T. simmonsii* showed neutral behaviour with the plant-promoting fungi *A. piperis* and *P. oxalicum*, on the other hand, *Trichoderma* sp. EXF-17020 showed antagonistic behaviour against both (e.g., the sub-type of growth above, Appendix A). Moreover, the strain EXF-17016 of *Trichoderma* sp. showed neutral behaviour with *A. piperis*, such as *T. simmonsii*, but antagonistic behaviour against *P. oxalicum*, such as the EXF-17020 strain. These findings demonstrate that fungal-fungal interactions are strain-dependent.

## 4. Conclusions

Biocontrol strategies are gaining a lot of attention and are expected to be applied more frequently in the agricultural sector in the near future. Among non-pathogenic microorganisms, *Trichoderma* strains are already used in green technologies for agriculture. According to our results, they indeed appear to be very effective against plant-pathogenic fungi. However, we have shown that they can also exhibit antagonistic behaviour against themselves and other antagonists of plant-pathogenic fungi. In addition, the type and strength of the behaviour is highly dependent on the species of the *Trichoderma* used. Although the co-cultures in this study were carried out under laboratory conditions and microbial behaviour in the open field also depends on other abiotic and biotic factors, such research offers a preliminary, yet rapid way to examine and predict how *Trichoderma* fungi will interact with other members of the soil microflora. Understanding the impact of biocontrol on soil microbial communities could be part of the new management practices in order to improve strategies for the biological control of pathogenic strains. In addition, the use of native *Trichoderma* strains as biological agents could lead to the preservation of soil microbial communities. All together, our findings offer a quantitative and easily translatable strategy for studying complex interactions with other fungi. This aspect has a substantial impact on the soil microbiome.

## Figures and Tables

**Figure 1 jof-09-00461-f001:**
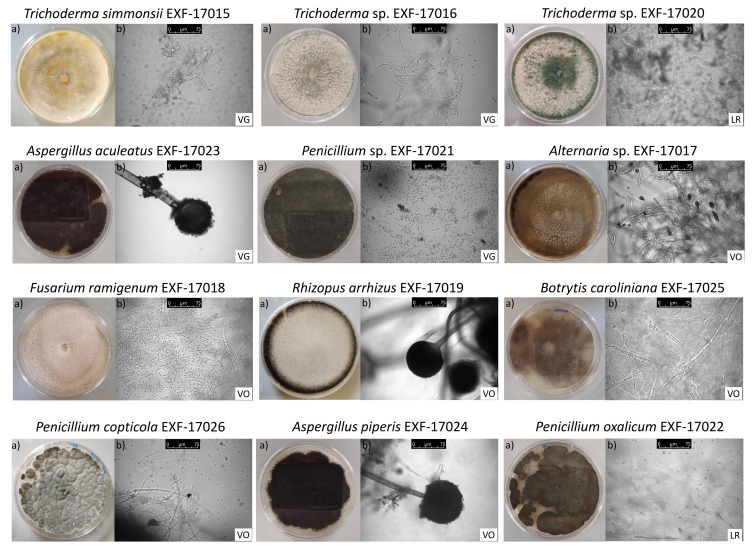
(**a**) Mycelial growth of fungal colonies on Petri dishes on a PDA medium of the twelve isolated plant-associated fungi from the three vineyards. (**b**) Microscopic observations of the mycelium and the reproductive spores of the isolated fungi. Scale bar 0–75 μm (Magnification 400×). Abbreviations: VG—Vigna Grande; VO—Vigne Olcru; LR—La Raia.

**Figure 2 jof-09-00461-f002:**
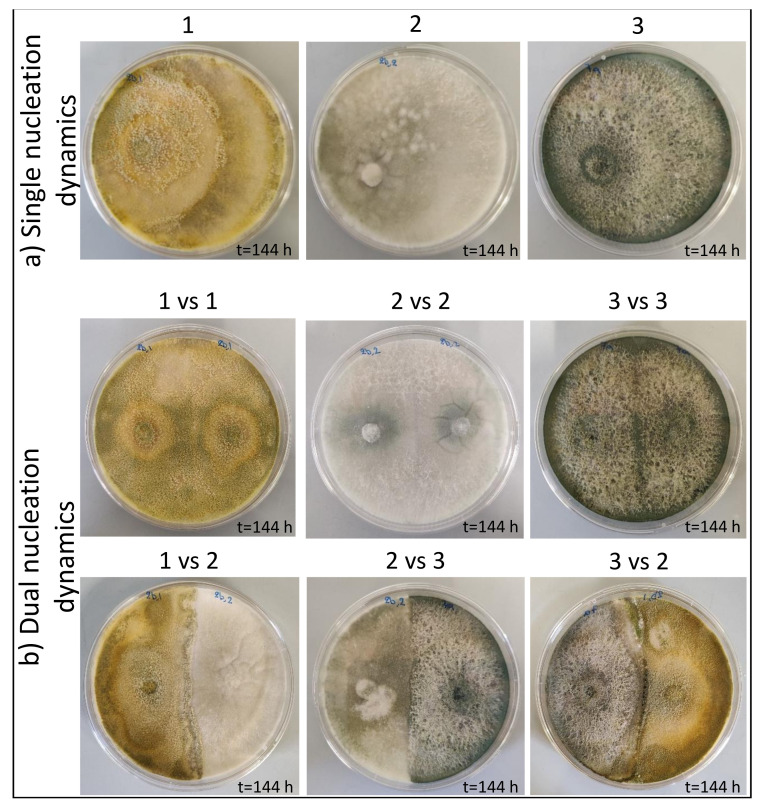
(**a**) Single nucleation dynamic and (**b**) Dual nucleation dynamic assay of *Trichoderma* strains. Fungal cultures were performed on a PDA medium as described in the Materials and Methods section. Pictures were taken up to 144 h post-cultivation. Fungal strains are as reported in Table 2, i.e., *Trichoderma simmonsii* EXF-17015 (No. 1), *Trichoderma* sp. EXF-17016 (No. 2), and *Trichoderma* sp. EXF-17020 (No. 3).

**Figure 3 jof-09-00461-f003:**
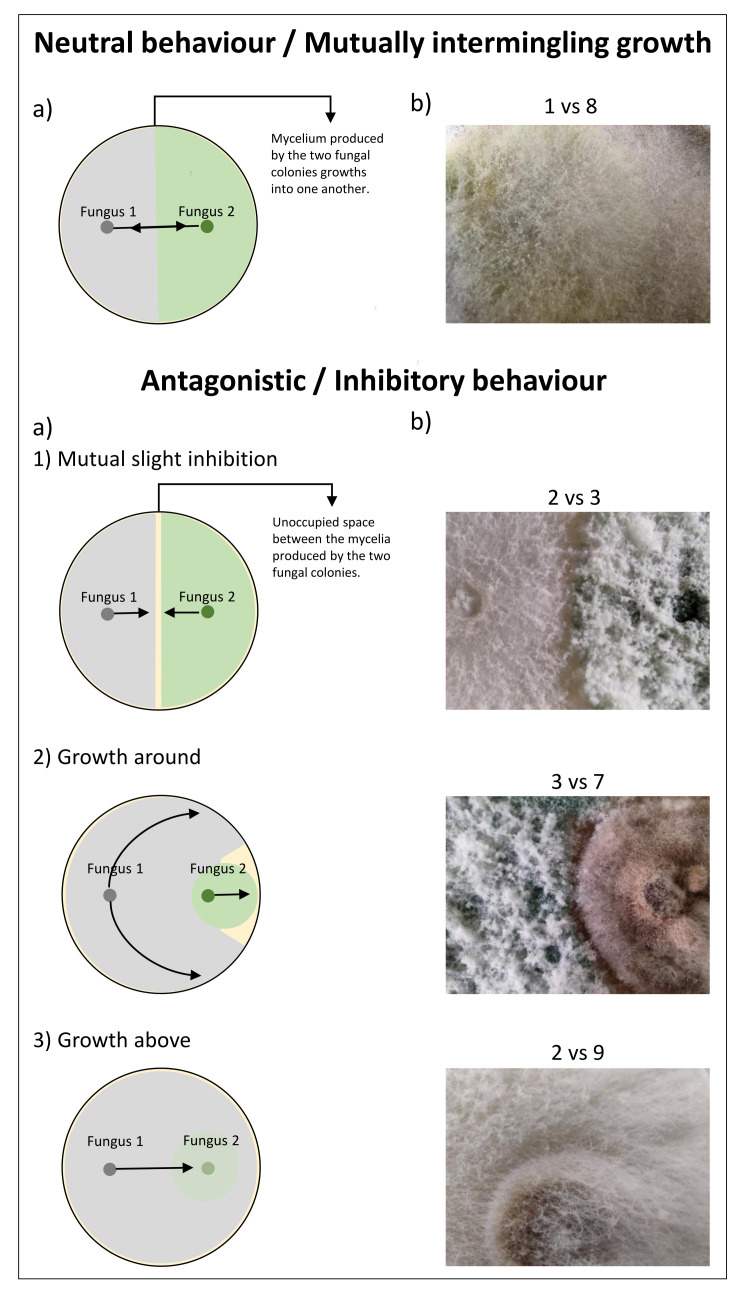
(**a**) A simplified schematic illustration of the interaction between fungi at the interface during dual nucleation assay. Arrows indicate the growth of the new-born mycelium produced by fungus 1 (i.e., *Trichoderma* strains) and fungus 2, and circles indicate the initial mycelial plugs, which served as inoculum (see Materials and Methods section for more details). ( **b**) Pictures at the interface of the dual nucleation dynamic assay of different fungi reported in Table 2, i.e., *Trichoderma simmonsii* EXF-17015 (No. 1), *Trichoderma* sp. EXF-17016 (No. 2), *Trichoderma* sp. EXF-17020 (No. 3), *Fusarium ramigenum* EXF-17018 (No. 7), *Rhizopus arrhizus* EXF-17019 (No. 8), and *Botrytis caroliniana* EXF-17025 (No. 9) (Magnification: 5X). Fungal cultures were performed on a PDA medium, as reported in the Materials and Methods section. Pictures were taken up to 144 h post-cultivation.

**Table 1 jof-09-00461-t001:** Location and date of sampling of the ten environmental samples used to isolate the plant-associated fungi.

No	Sample	Date of Sampling	Vineyard/Location
1	Dry leaves	17/01/2022	Vigna Grande/Brescia
2	Soil	17/01/2022	Vigna Grande/Brescia
3	Dry leaves	01/12/2021	Vigne Olcru/Pavia
4	Dry leaves	01/12/2021	Vigne Olcru/Pavia
5	Soil	01/12/2021	Vigne Olcru /Pavia
6	Dry leaves	01/12/2021	Vigne Olcru/Pavia
7	Dry leaves	02/05/2022	La Raia/Alessandria
8	Dry leaves	01/12/2021	Vigne Olcru/Pavia
9	Soil	17/01/2022	Vigna Grande/Brescia
10	Dry leaves	02/05/2022	La Raia/Alessandria

**Table 2 jof-09-00461-t002:** Isolated plant-associated fungal strains with their EXF number (Ex Culture Collection) and in which vineyards are found.

No.	Fungal Strain	Origin	Vineyard
1	*Trichoderma simmonsii* EXF-17015	Soil	Vigna Grande
2	*Trichoderma* sp. EXF-17016	Soil	Vigna Grande
3	*Trichoderma* sp. EXF-17020	Dry leaves	La Raia
4	*Aspergillus aculeatus* EXF-17023	Soil	Vigna Grande
5	*Penicillium* sp. EXF-17021	Dry leaves	Vigna Grande
6	*Alternaria* sp. EXF-17017	Dry leaves	Vigne Olcru
7	*Fusarium ramigenum* EXF-17018	Dry leaves	Vigne Olcru
8	*Rhizopus arrhizus* EXF-17019	Dry leaves	Vigne Olcru
9	*Botrytis caroliniana* EXF-17025	Dry leaves	Vigne Olcru
10	*Penicillium copticola* EXF-17026	Dry leaves	Vigne Olcru
11	*Aspergillus piperis* EXF-17024	Soil	Vigne Olcru
12	*Penicillium oxalicum* EXF-17022	Dry leaves	La Raia

## Data Availability

All fungal strains have been deposited in the Microbial Culture Collection Ex (Infrastructural Centre (IC) Mycosmo, Department of Biology, University of Ljubljana, Ljubljana, Slovenia) and are publicly available at https://www.ex-genebank.com, accessed on 4 April 2023.

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
