# Peer review of "A Pipeline to Investigate Fungal–Fungal Interactions: Trichoderma Isolates against Plant-Associated Fungi"

_jof, 2023, doi:10.3390/jof9040461_

Round 1

Reviewer 1 Report

Authors investigated the interaction of Trichoderma with other plant-associated fungi using a dual nucleation dynamic assay, which offers a pipeline to investigate the dual effect of the coexistence of different fungal strains. This manuscript was well organized, the figure quality is acceptable, which can be accepted but need some minor revisions.

1.       The title is not attractive, it is better to modify it.

2.       Line 20, keywords should be concise, no more than five words.

3.       The bio-mechanism of Trichoderma on plant pathogens should introduced in Introduction, https://doi.org/10.1016/j.tifs.2022.04.002 related references can be checked.

4.       Figure 2 and 3 could be merged into one figure. Another question, Figure 3, why use the Trichoderma fungi to grow against itself?

5.       In section of conclusion, in lab co-culture assay could access the antagonistic ability of Trichoderma on plant fungi via a fast manner, while the real situation in field would be different, this issue should be mentioned and discussed.

Author Response

We want to thank both reviewers for their comments and recommendations.

Following our reply on the revised version of our manuscript point by point.

Reviewer 1: The title is not attractive; it is better to modify it.

Authors: The title has been modified accordingly.

Reviewer 1: Line 20, keywords should be concise, no more than five words.

Authors: Keywords have been revised accordingly (p. 1, line 20).

Reviewer 1: The bio-mechanism of Trichoderma on plant pathogens should introduced in Introduction, https://doi.org/10.1016/j.tifs.2022.04.002 related references can be checked.

Authors:  A new paragraph describing the direct and indirect mechanisms by which Trichoderma fungi affect plant pathogens has been added to the revised version of the manuscript. Please consider p. 2, lines 38-60. Two new references have been also added (No. 9 and 10).

Reviewer 1: Figure 2 and 3 could be merged into one figure.

Authors: Figure 2 and 3 are now merged accordingly.

Reviewer 1: Another question, Figure 3, why use the Trichoderma fungi to grow against itself?

Authors: As stated on p. 9, lines 342-348, and shown in the new merged Figure 2, Trichoderma fungi also interact with each other. Moreover, we show that they do not harm each other. This is important, because the use of Trichoderma-based fungicides in an agricultural area will affect native Trichoderma strains in addition to other plant-associated fungi. We already discussed this aspect in the results and discussion and conclusion.

Reviewer 1: In section of conclusion, in lab co-culture assay could access the antagonistic ability of Trichoderma on plant fungi via a fast manner, while the real situation in field would be different, this issue should be mentioned and discussed.

Authors: The section Conclusions has been now revised accordingly (p. 12, lines 410-413).

Reviewer 2 Report

1-Molecular identification results should be improved and dendrograms drawn

2- The quality of Figure 1 is poor

3- Figure 1 does not have a scale bar

Author Response

We want to thank both reviewers for their comments and recommendations.

Following our reply on the revised version of our manuscript point by point

Reviewer 2: Molecular identification results should be improved and dendrograms drawn.

Authors: This point is beyond the purpose of the present paper.

Reviewer 2: The quality of Figure 1 is poor.

Authors: Figure 1 has now improved accordingly

Reviewer 2: Figure 1 does not have a scale bar.

Authors: The scale bar was present but we have now changed the color bar so it should now be clearer

Round 2

Reviewer 2 Report

Unfortunately, none of the requested items have been corrected

Author Response

We thank the review for his/her comments. We have answered to his/her points accordingly.  

Reviewer 2: Molecular identification results should be improved and dendrograms drawn.

Authors: This aspect was beyond the scope of our work. We 

Reviewer 2: The quality of Figure 1 is poor.

Authors: Figure 1 has now improved accordingly  (new Fig.1 added)

Reviewer 2: Figure 1 does not have a scale bar.

Authors: The scale bar was present but we have now changed the color bar so it should now be clearer (New Fig. 1 added)